# Definitions of Incidental [^18^F]FDG PET/CT Findings in the Literature: A Systematic Review and Definition Proposal

**DOI:** 10.3390/diagnostics14232764

**Published:** 2024-12-09

**Authors:** Jacob Pilegaard Mølstrøm, Natascha Lange, Manan Pareek, Anders Thomassen, Anne Lerberg Nielsen, Poul Flemming Høilund-Carlsen, Christian Godballe, Max Rohde

**Affiliations:** 1Research Unit for ORL—Head & Neck Surgery and Audiology, Odense University Hospital, J.B. Winsløws Vej 4, 5000 Odense, Denmark; cgodballe@health.sdu.dk (C.G.); max.rohde@rsyd.dk (M.R.); 2Faculty of Health Sciences, Department of Clinical Research, University of Southern Denmark, J.B. Winsløws Vej 19, 5000 Odense, Denmark; nataschalange@hotmail.com (N.L.); pfhc@rsyd.dk (P.F.H.-C.); 3Center for Translational Cardiology and Pragmatic Randomized Trials, Department of Biomedical Sciences, Faculty of Health and Medical Sciences, University of Copenhagen, 2200 Copenhagen, Denmark; mananpareek@dadlnet.dk; 4Department of Nuclear Medicine, Odense University Hospital, J.B. Winsløws Vej 4, 5000 Odense, Denmark; anders.thomassen@rsyd.dk (A.T.); anne.l.nielsen@rsyd.dk (A.L.N.)

**Keywords:** incidental, [^18^F]FDG PET/CT, finding, definition

## Abstract

**Objectives**: The objectives of this study were (1) to systematically review the currently used definitions of incidental 2-deoxy-2-[^18^F]fluoro-D-glucose positron emission tomography/computed tomography findings (IPFs) in the literature and (2) to propose an IPF definition. **Methods**: A systematic search was conducted according to the Preferred Reporting Items for Systematic Reviews and Meta-Analyses statement. The search was guided by the question “How is IPF defined?” and was performed in MEDLINE, Embase, and the Cochrane Library. The retrieved studies were reviewed and analyzed. The definitions of IPFs in the included studies were compiled into two sets of categories based on the description of *FDG uptake* and the specification of *clinical factors* in defining IPFs. **Results**: The systematic literature search identified 4852 publications accessible for title–abstract screening, which yielded 395 studies for full-text assessment. Sixty-five studies met the eligibility criteria and were included. Sixty-two percent mentioned “FDG uptake” in their definition. In 40% of the definitions, “Focal FDG uptake” was specified, while “FDG uptake in the surrounding tissue” was included in 15%. Fifty-seven percent stated that IPFs were “Unrelated to PET/CT indication”. Thirty-four percent specified IPFs as “Present in other organ than PET/CT indication”, whereas 20% included “No known disease related to IPF”. Seventeen percent of the definitions comprised a “New finding”, while 15% and 11% encompassed a “Clinical asymptomatic patient” and “Not a metastasis”, respectively. Finally, 5% of the definitions included “Potential clinical significance”. **Conclusions**: No generally accepted definition of IPFs currently exists. We propose an IPF definition based on explicit FDG uptake and clinical patient-related factors.

## 1. Introduction

Incidental findings are often encountered during 2-deoxy-2-[^18^F]fluoro-D-glucose positron emission tomography/computed tomography ([^18^F]FDG PET/CT). Previous studies have reported an overall prevalence of incidental PET/CT findings (IPFs) ranging from 7% to 12% [1,2]. However, the incidence, clinical relevance, and malignancy risk associated with IPFs vary significantly among different organ sites [3]. The most common sites are the thyroid, colon and rectum, prostate, and parotid glands [4]. The malignancy risk associated with IPFs varies considerably by anatomical site: approximately 10% in the parotid glands [5], compared with nearly 60% in both the breast and prostate [6,7].

Although numerous studies have investigated the incidence and clinical implications of IPFs, the reported outcomes have varied considerably [4]. An important reason may be that no universally accepted definition for IPFs exists. Nuclear physicians may tend to define IPFs according to the pattern of FDG uptake, i.e., focal or diffuse. Diffuse incidental uptake is generally considered to be associated with inflammation or physiological tracer uptake [8,9], while focal incidental FDG uptake is more likely to represent clinically significant findings, such as benign, pre-malignant, or malignant diseases [10]. Clinicians may account for other patient-related factors in their decision-making: Is a finding clinically relevant? Is there known prior patient disease related to the PET/CT abnormality? Is it a new finding or has it been described in previous imaging? Is the patient asymptomatic in relation to the finding?

The consequences of IPFs for the healthcare system are significant, including possible higher workloads and financial costs [11,12]. Moreover, from a scientific and epidemiological perspective, a clear definition of IPFs is required for the proper reporting and comparison of study results. The objectives of this study were (1) to systematically review the currently used definitions of IPFs in the literature and (2) to propose a definition of IPFs.

## 2. Materials and Methods

The protocol was registered in PROSPERO [13]. The systematic review was reported according to the Preferred Reporting Items for Systematic Reviews and Meta-Analyses (PRISMA) statement [14]. Inclusion and exclusion criteria were defined according to the PICOS (Patient, Intervention, Comparison, Outcome, and Study Design) criteria.

### 2.1. Search Strategy

The following electronic databases were searched: Ovid MEDLINE, Ovid Embase, and the Cochrane Central Register of Controlled Trials (CENTRAL) in the Cochrane Library on 5 September 2024.

The search strategy was established in collaboration with a research librarian from the University of Southern Denmark. It was developed for Ovid MEDLINE and then adapted for the other databases. No restrictions on the study design were applied. The limitations included only English, Danish, Norwegian, and Swedish publications from 2010 to 5 September 2024. The search was also restricted to human studies. A broad search was performed using search terms such as “Positron Emission Tomography” OR “FDG-PET/CT” AND “Incidental Findings” [Mesh] OR “Synchronous cancer”. To identify additional studies, reference lists of included trials were scrutinized. ClinicalTrials.gov and the WHO International Clinical Trials Registry Platform were searched for additional eligible, ongoing, or unpublished studies.

### 2.2. Selection of Studies

Three review authors (J.M., N.L., and M.R.) independently screened the titles, abstracts, and full-text manuscripts for inclusion using the Covidence systematic review software, Veritas Health Innovation, Melbourne, Australia. Available at www.covidence.org. The access date was 5 September 2024. Any inconsistencies or disagreements were discussed, and if consensus could not be reached, an additional review author was involved (C.G.). We included all randomized controlled trials, prospective and retrospective cohorts, reviews, and case–control studies, including experimental and observational studies. Case reports and conference abstracts were excluded.

### 2.3. Data Extraction and Management

Three review authors (J.M., N.L., and M.R.) extracted data from the included studies into a standardized Excel template. For each study, the data extraction included the title, first author, publication year, country, journal name, population, setting, and site of IPF. In the final analysis, only clear definitions of IPFs were included. We included only studies with clear definitions of incidental findings in combined [^18^F]FDG PET/CT scans.

### 2.4. Methodological Quality Assessment

Relevant quality-assessment tools were sought. However, the available assessment tools consider general study qualities, such as study design, statistical methods, etc. The Centre for Evidence-Based Medicine at Odense University Hospital was consulted and deemed that quality assessment of the included studies was unsuitable, as the definitions of IPFs were independent of the study designs, statistical methods, etc.

### 2.5. Data Synthesis

The definitions of IPFs in the included studies were compiled into two sets of categories, i.e., *Wording 1* and *Wording 2*. *Wording 1* comprised four categories based on the description of FDG uptake in defining IPFs whereas *Wording 2* contained seven categories based on the specification of clinical factors in defining IPFs.

### 2.6. Statistics

Simple arithmetic statistics were performed for the calculation of counts and percentages. For the analysis of wording frequencies, an online word counter was used (https://wordcounttools.com). The access date was 8 October 2024. 

## 3. Results

### 3.1. Search and Study Selection

The systematic literature search from 2010 to 5 September 2024 resulted in a total of 6320 studies. After the removal of 1468 duplicates, 4852 studies remained for title and abstract screening. In 395 of these cases, the full-text manuscript was assessed. Sixty-five studies met the eligibility criteria and were included for qualitative analysis. A PRISMA flowchart diagram of the study selection process is provided in Figure 1.

### 3.2. Study Characteristics

The characteristics of the 65 included studies [4,5,6,11,15,16,17,18,19,20,21,22,23,24,25,26,27,28,29,30,31,32,33,34,35,36,37,38,39,40,41,42,43,44,45,46,47,48,49,50,51,52,53,54,55,56,57,58,59,60,61,62,63,64,65,66,67,68,69,70,71,72,73,74,75] are presented in Table 1, including study design, country, and organ site with IPF.

### 3.3. Synthesis of Results

First, the 65 definitions were gathered in *Wording 1*, with four categories based on the specification of FDG uptake (Table 2).

Sixty-two percent mentioned “FDG uptake” in their definitions, whereas 38% did not describe FDG uptake as part of their definitions. In 40% of the definitions, “Focal FDG uptake” was specified, while “FDG uptake in the surrounding tissue” was included in 15%. Secondly, the 65 definitions were composed into *Wording 2*, with seven categories based on the phrasing of clinical features regarding IPF definitions (Table 3).

More than half of the definitions (57%) stated that IPFs were “Unrelated to PET/CT indication”. Roughly one-third (34%) specified IPFs as “Present in other organ than PET/CT indication”. Twenty percent of the definitions included the phrasing “No known disease related to IPF”. Furthermore, 17% of the definitions comprised “New finding” (i.e., a finding not seen on prior imaging), whereas 15% and 11% encompassed “Clinical asymptomatic patient” and “Not a metastasis”, respectively. Finally, 5% of the definitions included “Potential clinical significance”.

In Appendix A, an overview of the IPF definitions from the included studies and their compliance with the specific *Wording 1 and 2* criteria is listed.

## 4. Discussion

This is the first systematic literature review to have evaluated the currently used IPF definitions in the published literature. Our primary finding was that no recognized definition exists. We identified 65 different IPF definitions, all without any consensus criteria or scientific basis.

Although many studies have examined and published results concerning IPFs, most did not include clear definitions. This literature review yielded 395 studies investigating IPFs, but 215 of these (54%) were excluded, as they lacked a definition for IPFs. Many authors may consider IPFs a concrete, well-defined entity with sparse options for interpretation. Nonetheless, we found 65 studies that included IPF definitions, all of which were described uniquely.

The lack of a uniform IPF definition may lead to various classifications of IPFs, thereby diminishing the external validity of study results and challenging the traditional perception of IPF diagnosis. Indeed, reported incidence rates of IPFs have varied considerably, even within the same organ sites [1,4]. A clear definition would facilitate appropriate reporting and comparison of data from different settings. Based on this systematic review, it appears that both FDG uptake and several clinical patient-related factors should be taken into account.

[^18^F]FDG PET/CT naturally presents clinical challenges due to its highly sensitive capabilities. In most cases, IPFs are evident to nuclear physicians, but a final diagnosis cannot be established by imaging alone. For example, in Figure 2, an FDG-avid nodule appears in the left thyroid lobe but was proven benign by fine-needle aspiration.

The simplest form of the IPF definition contains no limitations in its wording but may often fall short in cases where different interpretations are made among nuclear physicians or clinicians. In Figure 3, a pulmonary infiltrate is suggestive of malignancy based on the CT alone. The PET/CT, on the other hand, suggests a benign etiology, given a low FDG avidity. The infiltrate was later proven benign due to complete regression on follow-up CT scans.

The variation in the IPF definitions found in the literature was somewhat expected. Some studies emphasized clinical findings, whereas others focused solely on imaging findings. Many of the included definitions were relatively similar and only differed in the order of words and use of various synonyms. Almost two-thirds of the studies characterized FDG uptake as part of their definitions.

Moreover, some definitions were rather descriptive, whereas others were simpler. For instance, Wan et al. [34] used a descriptive clinical definition: “Incidental findings, the outcome of interest, were defined as observations of potential clinical significance that were discovered and unrelated to the purpose or beyond the aims of the research studies in healthy, asymptomatic subjects or symptomatic patients with seemingly unsuspecting symptoms”. In contrast, Schaaf et al. [16] used a concise clinical definition: “Unexpected findings that are discovered when imaging is ordered for a completely unrelated cause”.

Confusion as to when an FDG-avid lesion represents an IPF is critical for proper patient guidance, decision-making, and healthcare planning—specifically, determining when and how to clinically intervene and when not to. Description of incidental findings may be beneficial; perhaps early-detected and therefore treatable cancer could be diagnosed. However, for some patients, incidental findings may lead to negative consequences such as anxiety, complications from (sometimes inappropriate) diagnostic procedures, or postponed cancer treatment [12,76]. The inconsistency in how and when nuclear medicine physicians report incidental findings may affect both the prevalence and recommended actions concerning IPFs. This underlines the importance of consensus building for both proper clinical practice and scientific study reporting.

The determination of IPFs will always depend, to some extent, on subjective clinical decision-making. Still, the current discrepancy concerning such an important clinical matter should not be ignored. Other significant consequences of IPFs for the healthcare system are also evident. IPFs constitute an increasing burden for clinical and paraclinical departments, with increasing workloads and possibly increasing financial costs [11,77]. For all these reasons, the establishment of consensus is central for achieving accurate and consistent IPF diagnoses. We suggest that the definition of IPFs should be based on explicit FDG uptake combined with specific clinical patient-related factors, such as indication, organ presentation, prior disease, and prior imaging. As such, we propose the following definition of IPFs: “*Focal FDG uptake unrelated to the PET/CT indication, and unknown from patient history and symptoms, including all previous examinations*”.

### 4.1. Limitations

Although this review is comprehensive, we may have missed a few definitions during our examination of the 395 full-text articles. IPF definitions are a challenging topic for a systematic review, given that the relevant studies are difficult to identify. We performed this comprehensive and careful literature review (including the body texts of the screened studies), involving 4852 studies, to identify all reported IPF definitions. We found 65 definitions of IPFs, all of which differed from one another. Furthermore, we excluded 215 studies that addressed IPFs but lacked clear definitions. While we believe these findings adequately cover the relevant published literature, we understand that the formal statistical power to detect clinically meaningful differences between the various definitions was not present, and any such comparison would ideally require prospective, perhaps even randomized, studies considering subsequent examinations and prognosis. However, this was beyond the scope of the present systematic review.

Another limitation was the comparison of the definitions based on wording. It was obvious that some definitions were much alike and only differed in sentence structures or the use of different synonyms. For that reason, the reviewers made a considerable assessment of the definitions rather than just a technical one and grouped them into the two sets of categories (i.e., *Wording 1 and 2*). The categorical criteria in *Wording 1 and 2* were constructed after assessing the included study definitions, which could potentially also be a source of bias.

The limitations regarding our proposed IPF definition should also be addressed. Since all of the IPF definitions in our systematic literature review were without a solid scientific base, our suggested definition also lacks firm evidence. Our proposed definition is characterized by being reasonable and developed as a “mean proportional” of the various definitions found in the literature. In other words, our IPF definition is pragmatic without being all-inclusive. For example, in some cases, diffuse FDG uptake may have significant clinical relevance and should therefore be communicated to treating physicians. For example, for breast cancer, even diffuse weak FDG uptake can indicate cancer, while in other cases, weak uptake may indicate relatively inactive cancer, as demonstrated by Kwee et al. [78].

In other cases, FDG uptake in surrounding tissues may signal inflammation, while in yet other cases, it may represent metastasis, in which the amount of FDG uptake depends not only on the aggressiveness of the cancer but also on the genotypic and phenotypic changes that have occurred relative to the primary cancer. Thus, there are many factors to consider, which probably explains why a generally accepted definition has not yet been found.

The definition of IPFs will thus often be pragmatic and dependent on local clinical expertise and decision-making practices. Nevertheless, we hope that our contribution will help qualify the debate and contribute to the search for a standardized common definition.

### 4.2. Future Work

Continued work in accordance with our IPF definition proposal would be an important step toward standardizing the clinical diagnosis of IPFs. We plan to implement it in daily clinical practice at our institution to improve consensus among our leading nuclear physicians.

Moreover, we aim to analyze data regarding IPFs in a large prospective cohort of patients who have undergone [^18^F]FDG PET/CT scans due to suspected head and neck cancer. The proposed definition will form the basis of that future work.

Subsequently, we plan to validate the proposed definition in a prospective inter- and intra-observer study among nuclear physicians. That study will have a rigorous setup, collecting information according to Wording 1 and 2 for all patients with possible IPFs. Long-term clinical outcomes will also be part of the outcome measures.

## 5. Conclusions

No generally accepted definition of IPFs currently exists. Accurate and consistent IPF diagnosis is imperative to patient care and to ensure standardized endpoints in clinical research dealing with [^18^F]FDG PET/CT findings. We propose an IPF definition based on explicit FDG uptake combined with clinical patient-related factors.

## Figures and Tables

**Figure 1 diagnostics-14-02764-f001:**
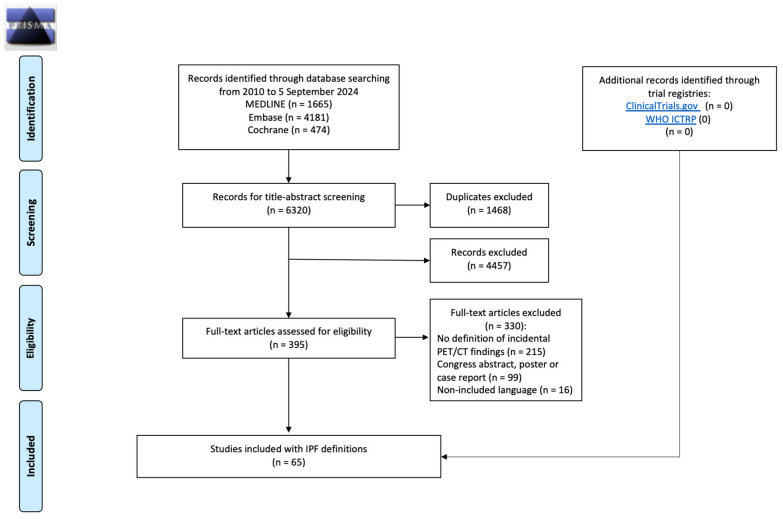
PRISMA 2009 flow diagram of the study selection process.

**Figure 2 diagnostics-14-02764-f002:**
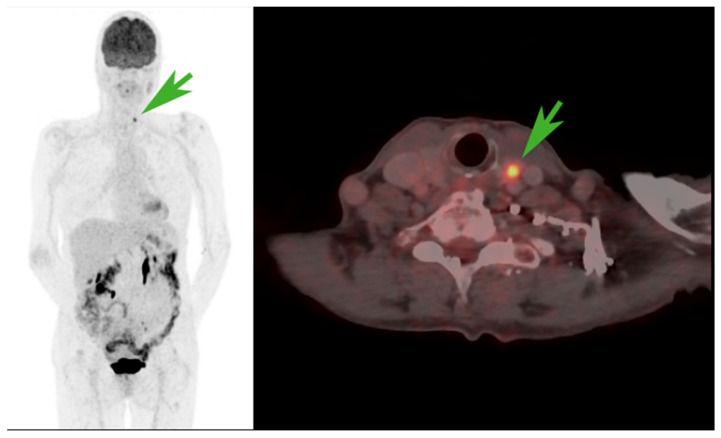
An incidental FDG-avid nodule (green arrow) in the left thyroid lobe that was later proven to be benign by fine-needle aspiration.

**Figure 3 diagnostics-14-02764-f003:**
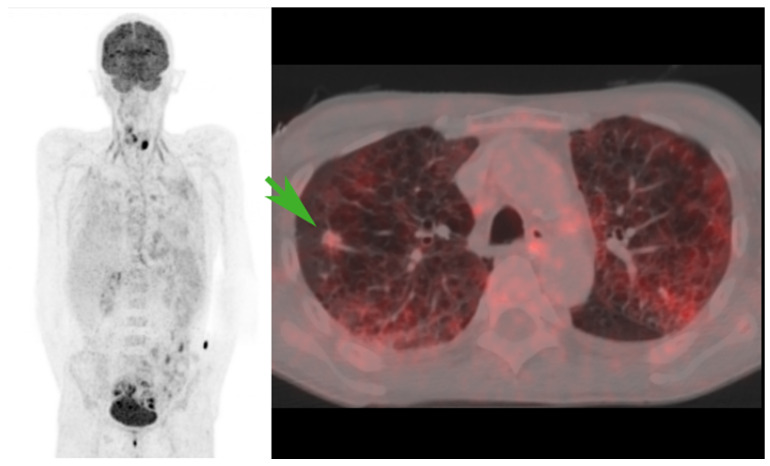
An incidental pulmonary infiltrate (green arrow) that was suggestive of malignancy by CT alone. [^18^F]FDG PET-CT, on the other hand, suggested a benign etiology, given a low FDG avidity. The infiltrate was later proven to be of a benign nature due to complete regression on follow-up CT scans.

**Table 1 diagnostics-14-02764-t001:** Study characteristics of the 65 included studies.

		*n*	%
Study design			
	Retrospective	47	72
	Review	12	18
	Prospective	5	8
	N/A	1	2
Country			
	Korea	11	17
	USA	8	12
	Italy	7	11
	Switzerland	5	8
	UK	5	8
	Canada	5	8
	China	5	8
	Spain	5	8
	Australia	4	6
	Denmark	2	3
	Columbia	1	1.5
	France	1	1.5
	India	1	1.5
	Romania	1	1.5
	Singapore	1	1.5
	Sweden	1	1.5
	Turkey	1	1.5
	Uruguay	1	1.5
Organ with IPF *			
	Thyroid	24	37
	Multiple organs	13	20
	Colorectal	11	17
	Prostate	6	9
	Parotid gland	4	6
	Breast	2	3
	Pharynx and larynx	2	3
	Pituitary gland	1	1.5
	Adrenal gland	1	1.5
	Gynecological	1	1.5

* IPF: incidental PET/CT finding.

**Table 2 diagnostics-14-02764-t002:** Wording 1: IPF definitions in the 65 included studies categorized according to description of FDG uptake.

Category	Level of Detail	*n* (%)
1	FDG uptake included	40 (62%)
2	Focal FDG uptake specified	26 (40%)
3	FDG uptake in the surrounding tissue included	10 (15%)
4	FDG uptake not described	25 (38%)

**Table 3 diagnostics-14-02764-t003:** Wording 2: IPF definitions in the 65 included studies categorized according to specification of clinical factors.

Category	Level of Detail *	*n* (%)
1	Unrelated to PET/CT indication	37 (57%)
2	Present in other organ than PET/CT indication	22 (34%)
3	No known disease related to finding	13 (20%)
4	New finding (not seen on prior imaging)	11 (17%)
5	Clinical asymptomatic patient	10 (15%)
6	Not a metastasis	7 (11%)
7	Potential clinical significance	3 (5%)

* The following terms were considered synonyms: unrelated; unexpected; new; incidentally; accidentally; unknown; without history; not associated.

## Data Availability

The original contributions presented in this study are included in the article/Appendix A. Further inquiries can be directed to the corresponding author.

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
