# Peer review of "Definitions of Incidental [18F]FDG PET/CT Findings in the Literature: A Systematic Review and Definition Proposal"

_diagnostics, 2024, doi:10.3390/diagnostics14232764_

Round 1

Reviewer 1 Report

Comments and Suggestions for Authors

The authors performed a systematic review of the literature about the definitions of incidental FDG PET/CT findings.

Through an accurate literature search (several databases were screened), study selection and data extraction, the authors demonstrated that no generally accepted definition of incidental FDG PET/CT findings currently exists. They also propose an incidental FDG PET finding definition based on explicit FDG uptake and clinical patient-related factors.

In my opinion, the topic of this article is timely and interesting for the readers of the journal. The methodology of the systematic review is adequate.

As minor revisions:

- please modify the title adding "[18F]FDG" as the review is focused on [18F]FDG PET only. The revised title should be: "Definitions of Incidental [18F]FDG PET/CT Findings in the Literature: A Systematic Review and Definition Proposal."

- please use the correct nomenclature for the radiopharmaceutical ([18F]FDG) in the title, main text, tables and figures.

Reviewer 2 Report

Comments and Suggestions for Authors

·         The frequency and clinical implications of incidental PET/CT findings (IPF) can vary considerably. In the present article, the authors aimed to systematically review currently used definitions of IPF in the literature and propose a definition of IPF for universal standardization.

·         Firstly, I would commend the authors as this is an interesting and unconventional topic for performing a systematic review. However, clinical utility of the same is limited. Most definitions, despite differences in wording and sentence structure, do convey the key central message that FDG uptake was incidental in nature, not related to the PET/CT indication and not typically expected based on patient symptoms and history.

·         The authors have proposed a IPF definition – “Focal FDG uptake unrelated to the PET/CT indication, and unknown from patient history and symptoms, including all previous examinations”. In several instances, IPFs may be in the form of diffuse FDG uptake and need not represent malignancy. For example, in thyroiditis or in cases with inflammatory bowel disorder (IBD). I would request clarity on why diffuse or other patterns of FDG uptake were excluded? Because as highlighted in the examples, these patterns of diffuse FDG uptake also have significant clinical relevance and must be communicated to the treating physicians.

·         The lack of formal statistical analysis and quality assessment also reduce scientific vigour of the present study.

·         Minor formatting issue. Line 127. Duplication of the period ‘.’ at the end of the table heading. “Table 2. Wording 1: IPF definitions in the 65 included studies categorized according to description of FDG uptake. .”

·         Minor formatting issue. Figure 1. Exclude the red-word mark (suggesting potential spelling error) under the word “Embase”.
